# Medical Nutrition Therapy and Physical Exercise for Acute and Chronic Hyperglycemic Patients with Sarcopenia

**DOI:** 10.3390/nu17030499

**Published:** 2025-01-29

**Authors:** Ángel Luis Abad-González, Silvia Veses, María Argente Pla, Miguel Civera, Katherine García-Malpartida, Carlos Sánchez, Ana Artero, Fiorella Palmas, Eva Perelló, Christian Salom, Ning Yun Wu Xiong, Clara Joaquim

**Affiliations:** 1Endocrinology and Nutrition Department, Hospital General Universitario Dr. Balmis, 03010 Alicante, Spain; angeluis1024@gmail.com; 2Instituto de Investigación Sanitaria y Biomédica de Alicante (ISABIAL), 03010 Alicante, Spain; 3Endocrinology and Nutrition Department, Hospital Universitario Doctor Peset, 46017 Valencia, Spain; svesesm@gmail.com (S.V.); kathe.garciamalpartida@gmail.com (K.G.-M.); chsaven@gmail.com (C.S.); 4Endocrinology Department, Hospital Universitari i Politècnic La Fe, 46026 Valencia, Spain; mariaargentepla@gmail.com; 5Endocrinology and Nutrition Department, University Clinical Hospital, Valencia, INCLIVA Biomedical Research Institute, 46010 Valencia, Spain; mi.civeraa@comv.es; 6School of Health Sciences, Universidad Cardenal Herrera-CEU, CEU Universities, Calle Grecia 31, 12006 Castellón, Spain; 7Endocrinology and Nutrition Department, Consorcio Hospital General Universitario de Valencia, Departamento de Medicina, University of Valencia, 46016 Valencia, Spain; carlos.sanchez@uv.es (C.S.); ana.artero@uv.es (A.A.); 8Endocrinology Department, Hospital Universitari Vall d’Hebron, 08035 Barcelona, Spain; fiorellaximena.palmas@vallhebron.cat; 9Endocrinology Department, Hospital Universitario San Juan de Alicante, 03550 Alicante, Spain; evapc89@hotmail.com; 10Endocrinology Department, Hospital Clínico Universitario de Valencia, 46010 Valencia, Spain; nywx.224@hotmail.com; 11Endocrinology Department, Hospital Universitari Germans Trias i Pujol, 08916 Badalona, Spain

**Keywords:** hyperglycemia, diabetes mellitus, sarcopenia, inflammation, aging, nutritional formulae, physical exercise

## Abstract

A wide range of factors contribute to the overlap of hyperglycemia—acute or chronic—and sarcopenia, as well as their associated adverse consequences, which can lead to impaired physical function, reduced quality of life, and increased mortality risk. These factors include malnutrition (both overnutrition and undernutrition) and low levels of physical activity. Hyperglycemia and sarcopenia are interconnected through a vicious cycle of events that mutually reinforce and worsen each other. To explore this association, our review compiles evidence on: (i) the impact of hyperglycemia on motor and muscle function, with a focus on the mechanisms underlying biochemical changes in the muscles of individuals with or at risk of diabetes and sarcopenia; (ii) the importance of the clinical assessment and control of sarcopenia under hyperglycemic conditions; and (iii) the potential benefits of medical nutrition therapy and increased physical activity as muscle-targeted treatments for this population. Based on the reviewed evidence, we conclude that a regular intake of key functional nutrients, together with structured and supervised resistance and/or aerobic physical activity, can help maintain euglycemia and improve muscle status in all patients with hyperglycemia and sarcopenia.

## 1. Introduction

The prevalence of diabetes is increasing among the elderly population, and it is estimated to be 25% among individuals aged 65 years and older [1].

Sarcopenia is emerging as a significant complication, particularly in individuals with type 2 diabetes mellitus (T2DM) [1,2,3]. The prevalence of sarcopenia is significantly higher in T2DM patients compared to the general population, ranging between 7% and 29.3%, depending on the diagnostic criteria. Moreover, it is known that diabetes accelerates sarcopenia through several mechanisms, including hyperglycemia, chronic inflammation, and oxidative stress (OS) [4]. Several risk factors associated with the co-occurrence of sarcopenia and T2DM have been identified, including advanced age, low body mass index (BMI), elevated values of glycosylated hemoglobin (Hba_1c_) or ultrasensitive C-reactive protein (CRP), longer duration of diabetes, and the presence of diabetic nephropathy [4,5].

Sarcopenia is associated with both functional and physical performance impairment, adversely affecting quality of life and increasing the risks of falls, fractures, hospitalizations, and even mortality. Despite these significant consequences, further research is needed to better define the measurement, diagnosis, and monitoring of sarcopenia [5].

The influence of nutrients on health outcomes exhibits age-dependent variations. As a result, dietary therapy strategies may need to shift with age from addressing obesity and metabolic syndrome to focusing on the prevention of frailty, particularly in diabetic patients aged 75 years and older who present with frailty, sarcopenia, or malnutrition [5,6].

The effects of diet and exercise on sarcopenia in patients with T2DM or acute hyperglycemia remain mostly underexplored. Regarding diet, a low intake of energy, protein, vitamin D, and omega-3 (ω-3) fatty acids has been associated with an increased risk of sarcopenia [4,7]. In terms of physical activity, although intervention studies targeting older and non-obese diabetic patients are limited, growing evidence reveals its benefits [8]. Particularly, resistance exercise has been shown to improve muscle mass and strength, while aerobic exercise enhances physical performance in individuals with sarcopenia [9,10].

This review emphasizes the need to explore the hyperglycemia–inflammation–sarcopenia axis and the impact of nutritional formulas and exercise on motor and muscle function in patients experiencing glycemic stress.

## 2. The Impact of Hyperglycemia on Motor and Muscle Function

### 2.1. Hyperglycemia and Muscle Function

Hyperglycemia is associated with the development of OS and chronic inflammation, which impact different organs and contribute to comorbidities, such as cardiomyopathy, nephropathy, neuropathy, periodontal disease, and skeletal muscle dysfunction [1]. Skeletal muscle atrophy, or its more severe form, sarcopenia, is notably higher in the presence of diabetes, and it is now recognized as another pathophysiological feature of diabetes, alongside traditional microvascular and macrovascular complications [2,3]. This muscle involvement makes diabetic patients more prone to developing sarcopenia and its progression. In addition to accelerated muscle mass decline, diabetic patients show a higher prevalence of dynapenia, which results in muscle weakness without an accompanying loss of muscle mass [5,6]. Potentially modifiable behavioral factors, such as nutritional status and physical activity levels, play a significant role in these processes [3,4,5].

Skeletal muscle mass progressively declines with age, contributing to increased insulin resistance (IR), reduced strength, physical limitations, and higher morbidity and mortality in the elderly population. These processes are often accompanied by a chronic proinflammatory state.

IR is a condition in which cells fail to adequately respond to insulin. This deficient insulin signaling is caused by various alterations, including rare mutations in the insulin receptor substrate (IRS), particularly in subtype 1, or post-translational modifications affecting the insulin receptor, IRS, or downstream effector molecules. The most common alterations in insulin resistance include a reduction in the number of insulin receptors and their catalytic activity, increased Ser/Thr phosphorylation of the insulin receptor and IRS, elevated tyrosine phosphatase activity—primarily of protein tyrosine phosphatase 1B (PTP-1B), which dephosphorylates the receptor and IRS—and decreased activity of phosphatidylinositol 3-kinase (PI3K), protein kinase B (Akt), and mammalian target of rapamycin (mTOR), crucial for muscle contractile protein synthesis. Simultaneously, IR upregulates protein degradation through the ubiquitin–proteasome system, resulting in reduced muscle mass. Additionally, defects in GLUT-4 expression and function further impair glucose uptake in muscle and adipose tissues, leading to significant metabolic disturbances [11].

Auto-phosphorylated residues on the insulin receptor are recognized by adaptor proteins, including members of the IRS family, with IRS-1 and IRS-2 serving as the primary intermediaries in the early stages of insulin signal transduction. IRS proteins act as adaptor molecules, organizing the formation of molecular complexes and initiating intracellular signaling cascades. Insulin’s actions are mediated primarily through the activation of two key signaling pathways: the PI3K/Akt Pathway, responsible for most of insulin’s metabolic effects, including glucose uptake and glycogen synthesis, and the MAPK/Ras Pathway, which regulates gene expression and mediates the mitogenic effects of insulin.

Research has identified an association between elevated HbA_1c_ values and the longer duration of diabetes with the appearance of sarcopenia. This relationship may be explained by the deterioration of metabolic control, which leads to increased proteolysis and reduced protein synthesis [7]. Comorbidities, such as diabetic nephropathy, particularly in its uremic phase, are additional risk factors for sarcopenia since it has been associated with decreased protein intake, increased proteolysis, and the presence of metabolic acidosis [8]. In this sense, chronic microvascular and macrovascular complications of T2DM also negatively impact muscle health: diabetic nephropathy can affect muscle mass, peripheral neuropathy can reduce muscle strength, and peripheral vascular disease can lead to muscle ischemia, as well as decreased muscle strength, muscle mass, and physical performance [9,10].

Chronic hyperglycemia contributes to the formation of advanced glycation products in skeletal muscle, which are, in turn, associated with reduced grip strength, gait speed, and endothelial dysfunction of intramuscular microcirculation [2,3].

All of this suggests that sarcopenia can now be considered a complication of T2DM, alongside the classic vascular and neurological complications. Furthermore, sarcopenia reduces tissue responsiveness to insulin, thereby increasing IR. This creates a vicious cycle that ultimately may result in the development and progression of T2DM (Figure 1).

### 2.2. The Hyperglycemia–Inflammation–Sarcopenia Axis

Aging is accompanied by several progressive changes in body composition, including reductions in muscle and bone mass and an increase in fat percentage, particularly visceral fat [2,12]. These changes contribute to heightened IR, diminished strength, physical limitations, and increased rates of morbidity and mortality in the elderly. A chronic proinflammatory state, coupled with potentially modifiable behavioral factors, such as nutritional status and physical activity levels, plays a significant role in driving these processes [13,14]. Inflammation, a protective biological response triggered by factors such as pathogens, damaged cells, and toxins, becomes problematic when it persists unchecked. Chronic inflammation and dysregulated inflammatory responses are closely associated with the pathophysiology of numerous chronic diseases [15,16].

Among elderly individuals, secretion of inflammatory cytokines and mitochondrial dysfunction have been identified as contributors to the development of sarcopenia, together with reduced physical activity, impaired nutrient absorption, and hormonal and neurovascular changes [17].

T2DM and sarcopenia share overlapping inflammatory mechanisms in their pathophysiology and are linked to increased free radical production and a reduction in antioxidant capacity [2,12,18]. Substantially increased glucose levels can lead to glucotoxicity, which promotes adipose tissue accumulation and activates processes such as protein glycosylation, OS, and inflammation [19,20]. In fact, chronic systemic inflammation plays a role in the development of sarcopenic obesity. The pro-inflammatory state associated with both obesity and aging has detrimental effects on skeletal muscle, inhibiting protein synthesis, reducing oxidative capacity, and exacerbating IR [21].

Low-grade inflammation is characterized by elevated levels of various proinflammatory molecules, including tumor necrosis factor-alpha (TNF-α), interleukins such as IL-1β, IL-6, and IL-1 receptor antagonist (IL-1ra), and CRP. These molecules are strongly linked to the development of pathological conditions, such as obesity, metabolic syndrome, and diabetes, where they play a critical role in IR, through different mechanisms [17,22]. For instance, skeletal muscle accounts for approximately 80% of glucose uptake under euglycemic hyperinsulinemia. As such, skeletal muscle IR is a pivotal factor in the development of T2DM and often precedes by decades pancreatic beta-cell failure and the onset of hyperglycemia [23].

Chronic inflammation disrupts insulin signaling, including a reduction in the synthesis of insulin-like growth factor-1 (IGF-1), leading to resistance to protein anabolism and increased proteolysis. Additionally, sustained hyperglycemia promotes the accumulation of advanced glycation end products (AGEs) in muscle and cartilage, resulting in muscle stiffness. This stiffness is associated with declines in grip strength and walking speed. Physical inactivity or disease-related malnutrition may further exacerbate this adverse metabolic context and accelerate the development of sarcopenia commonly observed in T2DM.

### 2.3. Hyperglycemia and Oxidative Stress

#### 2.3.1. Oxidative Stress and Free Radicals

OS arises from an imbalance between the production of free radicals and the capacity of antioxidant defenses, resulting in a shift toward an oxidative state. This state is characterized by the accumulation of reactive oxygen species (ROS) and reactive nitrogen species (RNS). They can originate from both endogenous and exogenous sources. Endogenous sources include nicotinamide adenine dinucleotide phosphate (NADPH) oxidase, myeloperoxidase (MPO), lipoxygenase, mitochondria, and xanthine oxidase. Exogenous sources encompass environmental factors, such as air and water pollution, tobacco use, alcohol consumption, exposure to heavy metals, certain drugs, industrial solvents, cooking contaminants, and radiation [24,25].

Under normal conditions, ROS and RNS play essential roles in metabolism, immune response, and cell proliferation and differentiation. However, under pathological conditions, an imbalance between antioxidant defenses and increased production of ROS and RNS can lead to oxidative damage to organelles, carbohydrates, proteins, nucleic acids, and lipids. Consequently, ROS and RNS can damage tissues and contribute to the development of various diseases, including sarcopenic obesity [26]. This damage progressively deteriorates muscle mass and disrupts muscle metabolic functions [27].

#### 2.3.2. Oxidative Stress in the Pathogenesis of Obesity, Sarcopenia and Type 2 Diabetes Mellitus

In elderly individuals, OS contributes to the development of sarcopenia and obesity through mechanisms such as mitochondrial dysfunction, endoplasmic reticulum (ER) stress, and imbalances in muscle mass regulation [28]. OS is a key factor linking sarcopenic obesity to its associated comorbidities [12]. It is also closely associated with other contributors to sarcopenic obesity, such as inflammation, hormonal imbalances, and behavioral factors [28].

In obesity, mitochondrial dysfunction is linked to oxidative damage to adipocytes and enlargement of adipocyte size, both of which exacerbate metabolic dysfunction [29,30]. Mitochondrial dysfunction caused by OS in obesity is a key factor underlying IR and muscle atrophy [31]. Age- and diabetes-related muscle loss has been associated with mitochondrial abnormalities in content, size, morphology, and function. Compared to younger individuals, older adults have a reduced oxidative capacity per unit of muscle, resulting in impaired mitochondrial function [27,32,33,34,35,36,37].

Similarly, nutrient overload in obesity can trigger ER stress, which is also a significant contributor to sarcopenia. ER stress is particularly elevated in the skeletal muscles of older individuals [38,39,40,41]. Maintaining muscle mass requires a delicate balance between anabolic and catabolic pathways. OS disrupts this balance by inhibiting anabolic pathways, such as the PI3K/Akt/mTOR pathway [42,43,44], while activating catabolic pathways, including the ubiquitin–proteasome and autophagic/lysosomal systems. These catabolic pathways degrade dysfunctional organelles and damaged proteins through lysosome-dependent processes involving muscle satellite cells [45,46].

With aging, abnormal autophagy in muscles leads to the accumulation of damaged mitochondria, increasing ROS production and further exacerbating ER stress, impaired turnover of sarcomeric proteins, and cell death. This cascade results in progressive muscle loss [47]. Additionally, the regenerative capacity of satellite cells—critical for muscle fiber maintenance, repair, and remodeling—declines with age and limits the ability to renew and regenerate muscle tissue [48,49,50].

These interconnected mechanisms perpetuate a cycle of lipo-toxicity, chronic inflammation and worsening IR, and accelerate muscle mass loss. This vicious cycle has harmful consequences for muscle health, ultimately impairing physical function and reducing independence in individuals suffering from sarcopenic obesity [51].

OS also impairs insulin signaling and disrupts glucose uptake in skeletal muscle, thereby contributing to hyperglycemia. OS plays a relevant role in both the onset and progression of diabetes and its complications [52]. Factors such as hyperglycemia, overproduction of free fatty acids, hypoxia, inflammation, and immune response lead to OS, which contributes to perpetuating hyperglycemia through IR, chronic inflammatory state, and endothelial dysfunction [52]. An illustrative overview of the relationship between OS, hyperglycemia, IR, and sarcopenic obesity is presented in Figure 2.

#### 2.3.3. Oxidative Stress and Aging

The oxidation–inflammatory theory of aging, also known as “oxy-inflammation-aging” suggests that aging is characterized by chronic OS, which activates the immune system and leads to a persistent inflammatory state. This creates a vicious cycle where OS and inflammation reinforce each other, causing progressive damage to structures, tissues, and organs [53]. In this scenario, OS is largely attributed to the significant oxygen consumption in skeletal muscles, which results in the excessive production of ROS and RNS species.

Among the many factors contributing to skeletal muscle dysfunction, OS driven by IR plays a central role. IR is a feature of both aging and obesity. Aging also leads to increased body fat, particularly in the abdominal region (visceral fat), as well as the accumulation of fat within skeletal muscles (myo-steatosis) and the liver (hepatic steatosis), all of which exacerbate IR [54,55].

Lipid-induced intramuscular inflammation is another key mechanism linking diabetes to muscle atrophy. Adipocyte hypertrophy, often observed in obesity, triggers a chronic systemic inflammatory state characterized by decreased levels of adiponectin and elevated levels of CRP, leptin, TNF-α, and IL-6. This contributes to muscle inflammation, leading to IR and muscle atrophy through protein degradation mediated by nuclear factor kappa B (NF-κB) and other signaling pathways [56].

Obesity and aging further exacerbate muscle loss through hormonal imbalances, including reductions in growth hormone (GH), testosterone, estrogen, IGF-1, and adiponectin, as well as increased levels of myostatin. Physical inactivity, commonly associated with both obesity and aging, also contributes to muscle deterioration by impairing the respiratory, osteoarticular, and neuromuscular systems, resulting in a decline in physical function. In older individuals, declines in the frequency and amplitude of GH peaks, along with reductions in IGF-1 and adiponectin levels, are well-documented [57].

Hormonal imbalances also affect sex hormones: testosterone in men and estrogen, progesterone, and adrenal- and ovarian-derived androgens in women [58,59]. Notably, T2DM is often linked to male hypogonadism, with both conditions promoting sarcopenia in men [60].

In aging-associated sarcopenia, neuromuscular fatigue during exercise and an age-related reduced response to physical training are prevalent [61]. This decline is closely tied to changes in the neuromuscular unit (NMU), which comprises three key components: presynaptic elements (motor neurons), intrasynaptic components (synaptic basal lamina), and postsynaptic elements (muscle fibers and muscle cell membranes). Aging reduces the nerve terminal area and the number of postsynaptic folds, leading to impaired postsynaptic responses [61]. Moreover, mitochondria show significant age-related reductions in number in the NMU and are often marked by high levels of oxidative damage, decrease in synaptic connections, and reduced neurotransmitter release during depolarization, all of which contribute to neuromuscular deterioration [62].

## 3. Importance of the Clinical Assessment of Sarcopenia in Hyperglycemia

The assessment and detection of sarcopenia in patients with hyperglycemia is of great interest due to the clear association between both conditions. Several epidemiological studies suggest that diabetes may accelerate the decline in both muscle mass and functionality, thereby increasing the risk of sarcopenia [63]. This association has been confirmed by meta-analyses, including those conducted by Anagnostis et al. and Chung et al. [13,64]. The first of these found that patients with T2DM have a significantly higher risk of sarcopenia (odds ratio [OR] 1.55; *p* < 0.001), mainly due to decreased muscle strength and performance, while no significant differences were seen in muscle mass [13]. The second meta-analysis included studies employing at least two of the three diagnostic criteria for sarcopenia (low muscle mass, reduced muscle strength, and/or impaired muscle performance) and revealed that T2DM patients had poorer muscle strength and performance compared to euglycemic individuals, while muscle mass also remained comparable between the groups [64].

Another meta-analysis by Veronese et al. found a higher prevalence of sarcopenia in patients with T2DM compared to control subjects (OR 1.63; *p* = 0.002) and suggested a potentially bidirectional relationship between the two conditions. Additionally, the prevalence of sarcopenia was significantly higher in T2DM patients with macrovascular and microvascular complications compared to those without such complications (OR 2.44; *p* < 0.0001) [65]. These results, together with evidence of decreased capillarization in skeletal muscle of patients with sarcopenia [66] and the involvement of AGEs in skeletal muscle atrophy and dysfunction [67], underscore a potential vascular role in the development of sarcopenia.

Efforts have been made to identify predictive risk factors for sarcopenia in patients with T2DM. Current findings show that older age, male gender, prolonged hyperglycemia, obesity, and osteoporosis are risk factors, while metformin administration and low BMI are associated with reduced risk of sarcopenia [68,69].

## 4. Specific Nutritional Formulas for Patients with Stress Hyperglycemia or Diabetes and Sarcopenia

Neither clinical trials nor observational studies have specifically addressed the effect of nutritional supplementation with specific formulas on sarcopenia in patients with T2DM or stress hyperglycemia. Therefore, conclusions and recommendations are based on studies not designed with this particular focus on the diabetic population, along with non-systematic reviews.

The American Diabetes Association (ADA) highlights the lack of a universally ideal distribution of macronutrients for patients with diabetes and that this should be individualized based on each patient’s needs [70]. ADA recommends increasing fiber intake to at least 14 g per 1000 kcal, with at least half of that coming from whole grains, as this is associated with lower all-cause mortality in individuals with T2DM [71]. There is no consensus on protein intake among ADA representatives. Moreover, it is generally accepted that carbohydrates should account for 45–60% of the total caloric intake, while lipids should make up 20–35%, with 7–10% of those lipids being monounsaturated fatty acids (MUFA) [72,73]. Nevertheless, these recommendations pertain to oral intake and not enteral formulas. Most standard enteral formulas are high in carbohydrates (typically over 50% of total calories), contain around 30% fat (with only 8% of that being MUFA), and usually lack fiber. As a result, they can contribute to higher postprandial hyperglycemia and hyperinsulinemia [74].

Specific nutritional formulas for patients with diabetes employ different strategies to achieve metabolic goals. These strategies may include increasing fiber intake, modifying the carbohydrate profile, increasing the percentage of fats—primarily through MUFA—or a combination of these approaches [74].

### 4.1. Proteins, Carbohydrates and Fiber

Aging reduces the ability to efficiently dispose of glucose in skeletal muscle cells. Consequently, older adults with sarcopenia and diabetes require careful assessment of their carbohydrate and protein intake needs.

A double-blind randomized clinical trial (RCT) concluded that participants who consumed a diabetes-specific formula—containing whey protein, medium-chain triglycerides (MCTs), a lower percentage of rapidly absorbed carbohydrates, and fiber—showed a statistically significant improvement in postprandial glycemia, and a non-significant reduction in insulin secretion, compared to patients receiving a standard formula. Notably, the study did not specify whether the participants had sarcopenia or malnutrition or not [75].

Mayr et al. demonstrated that administering a diabetes-specific, carbohydrate-modified oral nutritional supplement (ONS) for 12 weeks to 40 elderly, normal-weight T2DM patients with prior involuntary weight loss resulted in a reduced postprandial glycemic response and improved long-term glycemic control [76]. In another study, long-term enteral tube feeding with a diabetes-specific enteral formula—low in carbohydrates and enriched with monounsaturated fatty acids (MUFAs), fish oil, chromium, and antioxidants—was found to significantly decrease total insulin requirements and reduce the incidence of hypoglycemia in insulin-treated patients with T2DM and neurological disorders, compared to a standard diet. Furthermore, fasting and afternoon glucose levels were markedly lower and led to improved overall glycemic control. The formula under investigation was safe and well tolerated [77,78]. Lansink et al. also demonstrated an improvement in postprandial glycemia with the administration of a high-protein diabetes-specific formula in ambulant, non-hospitalized patients with T2DM [79]. Another observational retrospective study in an elderly diabetic population taking a specific hypercaloric and hyper-proteic supplement reported a reduction in the number of hospitalizations, hospital stays, and emergency department visits, although no changes in glycemic control or treatment were observed [80]. Likewise, a low carbohydrate formula reduced insulin requirements and glycemic variability in intensive care patients [81].

There is little evidence on the impact of protein concentration coming from clinical trials in this context. The results of the NuAge5 study suggested that adequate protein intake (>1 g/kg body weight) helps preserve muscle strength and function in elderly women with T2DM, although this effect was not significant in men. However, the study did not assess pre-existing sarcopenia [82].

Since T2DM is a risk factor for sarcopenia, Tamura et al. stated that an adequate protein intake (1–1.5 g/kg body weight) is essential in elderly patients with T2DM. This recommendation does not apply to patients with kidney disease [5].

Regarding protein quality, Adams et al. highlighted that whey protein, rich in branched-chain amino acids, positively affects glycemic control by improving insulin sensitivity and reducing postprandial glycemia. These benefits are attributed to whey’s rapid absorption, increased gastric emptying, and stimulation of gastric inhibitory polypeptide (GIP) and glucagon-like peptide-1 (GLP-1) secretion, as well as the inhibition that the latter exerts on the dipeptidyl peptidase IV (DPP-IV) complex. Therefore, incorporating whey protein into nutritional formulas could enhance anabolism and improve glycemic control [83].

A review also examined the effects of supplementing amino acids or their metabolites, such as hydroxymethyl butyrate, leucine, arginine, and glutamine, in patients with sarcopenia and diabetes or stress hyperglycemia and found that evidence is insufficient to recommend such supplementation [84]. Under normal glycemic conditions, 80% of glucose clearance occurs in muscle tissue but, in patients with sarcopenia, reduced muscle mass can contribute to T2DM development. Additionally, certain diabetes therapies can exacerbate muscle loss; for example, metformin induces autophagy in muscle cells, while insulin’s anabolic effects diminish with age. Weight loss, although essential in managing diabetes with IR and obesity, may further reduce muscle mass [85].

Despite the current scarcity of evidence, protein intake in diabetic and elderly populations should exceed that of the general population, due to the increased prevalence of sarcopenia. Nutritional formulas should ideally be high in protein, with whey protein favored due to its rapid absorption and superior anabolic effects.

Dietary fibers, which are non-starch polysaccharides, are known to help regulate blood glucose levels by preventing sharp spikes caused by carbohydrate consumption, and lower cholesterol levels. Therefore, it is essential to consider not only the total carbohydrate intake but also factors such as the type of sugar, the form, and composition of food, as well as cooking and processing methods, as all these elements significantly influence glycemic responses [70].

Fiber is known for its low glycemic index (GI) and is a key ingredient in all diabetes-specific formulas. It is often included in high or exclusive proportions of fermentable fiber. This type of fiber has been shown to improve both glycemic and lipid profiles and promote the production of short-chain fatty acids in the colon through fermentation. Additionally, it can reduce carbohydrate absorption by delaying gastric emptying and accelerating intestinal transit.

In 2006, Hofman et al. demonstrated that diabetes-specific enteral formulas have a lower GI compared to standard formulas. The study evaluated 12 enteral formulas, and the mean GI was 19.4 (range of 19 to 26) for diabetes-specific formulas and 42.1 (range of 25 to 61) for standard supplements. Nevertheless, half of the diabetes-specific feeds exhibited some overlap with the GI values of standard formulas [86].

Diabetes-specific formulas are typically composed of low-GI carbohydrates, including non-hydrolyzed starches, modified maltodextrins, polyols (e.g., maltitol), and disaccharides such as isomaltulose. The inclusion of fructose remains a topic of debate. While fructose offers high sweetening power, a low glycemic index, and insulin-independent cellular uptake, it has been postulated that high doses can lead to adverse effects, such as hypertriglyceridemia, elevated low-density lipoprotein (LDL) cholesterol, and IR. Contrary to these concerns, a certain study indicated that small doses of fructose (≤11 g per meal) could reduce the glycemic response to high-GI meals and lower postprandial glycemia by 4–30% [87].

Considering these findings and the higher prevalence of sarcopenia in diabetic patients, diabetes-specific, low-carbohydrate and high-protein formulas enriched with fermentable fiber should be prioritized over standard, normo-proteic formulas.

### 4.2. Diets High in Monounsaturated Fatty Acids and Antioxidants

#### 4.2.1. Monounsaturated Fatty Acids

The first meta-analysis to assess the effect of MUFAs in patients with diabetes was conducted by Elia et al. and included 23 studies with a total of 784 patients diagnosed with T1DM, T2DM, and stress hyperglycemia [88]. The study compared diabetes-specific formulas—high-fat content (40–50%), of which more than 60% of the fat was in the form of MUFAs, along with added fructose and fiber—to standard formulas. According to the analysis, diabetes-specific formulas produced a smaller postprandial increase in blood glucose concentrations than standard formulas (1.03 mmol/L, 95% CI: 0.58–1.47). Particularly, in six RCTs, the maximum blood glucose concentration was 1.59 mmol/L (95% CI: 0.86–2.32), and two RCTs showed a glucose area under the curve (AUC) of 7.96 mmol/L/min (95% CI: 2.25–13.66). In four studies, no significant effects on high-density lipoprotein (HDL), total cholesterol, or triglyceride levels were observed. One study reported a reduction of greater than 25% in insulin requirements and fewer diabetes-related complications with diabetes-specific formulas compared to standard formulas [84].

In 2014, Ojo et al. published a meta-analysis that specifically focused on patients with T2DM. This analysis evaluated the impact of diabetes-specific formulas, rich in low-GI carbohydrates and MUFAs, versus standard formulas on glucose and lipid parameters. Diabetes-specific formulas improved postprandial glycemia, HbA_1c,_ and insulinemic response compared to standard formulas [89].

A recent systematic review and meta-analysis examined studies comparing MUFA-rich diabetic enteral formulas with standard polymeric formulas, which respectively contained between 7.9% and 19.4% of MUFAs, in patients with diabetes or stress hyperglycemia. Eighteen studies with 845 patients were included, and the analysis found that the use of MUFAs was associated with improvements in postprandial outcomes, such as glycemic peak, AUC, insulin AUC, and mid-term follow-up parameters (HbA_1c_ and glycemic variability). Standardized mean differences were calculated for each parameter and statistically significant reductions in median postprandial glycemic peak (1.53, 95% CI: −2.44 to −0.61), incremental glucose response (1.19, 95% CI: 0.68 to 1.71), and insulin AUC (0.65, 95% CI: 0.26 to 1.03) were observed. The study also showed improvements in HbA_1c_ (0.63, 95% CI: −0.05 to −1.21), glycemic variability (0.93, 95% CI: −0.31 to −1.55), and a reduction in the mean insulin dose administered (0.49, 95% CI: −0.14 to −0.85). According to these results, diabetes-specific formulas containing 20% or more energy from MUFAs or 40% or more energy from fat have beneficial effects on peak postprandial glucose, incremental glucose response, and glucose variability compared to standard formulas [90]. These results have been questioned due to the heterogeneity of the included studies [91].

Recent advances in diabetes-specific formulas have led to the recommendation of high-MUFA formulas, considering those that provide more than 20% of the total energy from MUFAs [74]. These formulas currently deliver the highest MUFA content compared to other specialized formulas on the market [92]. In this sense, in a multicenter, randomized parallel-group study comparing MUFA-rich diabetic oral supplements at different doses (2 or 3 daily supplements), a higher dose resulted in greater improvements, including a more significant reduction in HbA_1c_ (0.98% vs. 0.60%) and better nutritional parameters (weight, BMI, fat mass, albumin, and prealbumin) [93]. Additionally, a recent prospective multicenter real-life study evaluated the efficacy of a hypercaloric and hyper-proteic enteral nutrition (EN) formula containing more than 20% MUFA in its total caloric content, with 112 patients over a 24-week follow-up period. The results revealed a significant decrease in the proportion of patients with malnutrition from 78.6% to 29.9% (*p* < 0.001), as measured by the subjective global assessment. Furthermore, nutritional parameters, such as weight, BMI, albumin, prealbumin, and transferrin, significantly improved. Blood glucose and HbA_1c_ also showed marked reductions at both 12 and 24 weeks (blood glucose: 155.9–139.0–133.9 mg/dL, *p* < 0.001; HbA_1c_: 7.7–7.3–7.1%, *p* < 0.001). No significant changes were observed in cholesterol, triglycerides, creatinine, or glomerular filtration rate. Thus, this study supports the long-term use of diabetes-specific formulas [94].

To our knowledge, only one study has specifically evaluated the potential cardiovascular risk associated with these formulas, although it could be hypothesized that the high lipidic content could potentially worsen cardiovascular outcomes. This prospective clinical trial examined the effects of exclusive EN rich in MUFA on glycemic control and cardiovascular risk biomarkers in elderly diabetics. The study found a significant improvement in HbA_1c_ (from 6.1 ± 0.1% to 5.8 ± 0.1%, *p* < 0.045), along with reductions in monocyte chemotactic protein-1 and soluble E-selectin (*p* < 0.05) [95].

#### 4.2.2. Antioxidants

Antioxidants play a crucial role in defending against ROS-induced toxicity. Antioxidants can be either endogenous or exogenous [96]. Endogenous antioxidants include enzymatic complexes, such as superoxide dismutase (SOD), catalase (CAT), and glutathione peroxidase (GSH-Px), as well as non-enzymatic molecules, like bilirubin and β-carotene. Exogenous antioxidants encompass compounds such as ascorbic acid (vitamin C), α-tocopherol (vitamin E), and phenolic antioxidants, including resveratrol, phenolic acids, flavonoids, selenium, zinc, and acetylcysteine.

Given the strong connection between OS, diabetes, and sarcopenia, antioxidant supplementation has been used as a therapeutic approach to mitigate OS. Commonly studied antioxidants include SOD/CAT/GSH-Px mimetics, vitamins A, C, and E, β-carotene, flavonoids, selenium, zinc, N-acetylcysteine (NAC), and CoQ10. However, the physiology of the REDOX system is complex, and no significant benefits of antioxidants have been described for either diabetes or complications. This is likely due to issues such as the poor solubility, permeability, and stability of these compounds [60,97].

The InCHIANTI study, a prospective investigation of the factors contributing to reduced mobility in old individuals, revealed that higher serum antioxidant levels (e.g., carotenoids) were associated with a reduced risk of disability and muscle strength loss [98]. Another study performed on diabetic felines reported lower SOD levels compared to non-diabetic counterparts, findings that align with data from humans. Reduced serum SOD concentrations are associated with increased glycated proteins, IR, and diabetic complications. Notably, an exclusive diabetic diet given to cats for eight weeks resulted in increased GSH-Px activity, potentially due to selenium supplementation, a key cofactor for the GSH-Px enzyme [99].

Moreover, one study reported that the administration of diabetic EN, supplemented or not with arginine, is associated with lower mortality, lower IR, and improved peritoneal macrophage function in obese diabetic rats with infectious stress, compared to the use of standard enteral formula, probably due to the production of TNFɑ and NO [100].

### 4.3. Benefits of Omega-3 Intake and the Omega-6/Omega-3 Ratio

Omega-3 fatty acids are a class of polyunsaturated fatty acids primarily composed of eicosapentaenoic acid (EPA) and docosahexaenoic acid (DHA), whose supplementation effects in diabetes remain inconclusive. Earlier studies suggested that omega-3 fatty acids supplementation can reduce triglycerides and elevate HDL cholesterol, although it appears to have minimal or no significant impact on glycemic control, including HbA_1c_, IR, or fasting glucose levels [101]. Moreover, this research found no evidence suggesting that the ω-3/ω-6 ratio is a determining factor in T2DM.

A recent meta-analysis demonstrated a reduction in fasting blood glucose and IR. The precise mechanisms underlying these effects remain unclear, but proposed mechanisms include alterations in mitochondrial bioenergetics and mitigation of endoplasmic reticulum stress [102]. Furthermore, a multicenter randomized crossover study compared postprandial responses after consumption of two diabetes-specific formulas, one enriched with slowly absorbed carbohydrates and omega-3, and a standard formula. The study revealed that the diabetes-specific formulas led to a significantly lower postprandial glycemic response compared to the standard formula. Notably, the formula enriched with slowly absorbed carbohydrates and omega-3 resulted in a reduced insulin and glucose curve while increasing GLP-1 concentrations. These findings highlight the importance of nutritional composition, particularly the inclusion of omega-3, in optimizing enteral formulas for patients with diabetes [103].

### 4.4. Specific Nutrients to Control Diabetes and Oxidative Stress

Resveratrol, a polyphenol found in plant-based foods, has been shown to effectively prevent muscle mass loss, reduce myofiber shrinkage, mitigate declines in muscle strength, and limit the excessive accumulation of muscle fat when administered with a high-fat diet in rats [21]. Tocotrienols and green tea polyphenols have similarly demonstrated benefits in animal models of obesity, including increases in muscle mass and mitochondrial enzyme activity. Antioxidant compounds, like selenium and vitamins E and C, have also been proposed for managing OS-related conditions, such as sarcopenia. Unfortunately, the physiology of the redox system is more complex than initially believed [60] and these antioxidants may paradoxically act as prooxidants under certain conditions, potentially increasing the risk of mortality.

Polyphenols, widely present in foods like cocoa, fruits, vegetables, extra virgin olive oil, wine, and tea, are metabolized by the intestinal microbiota into a diverse array of metabolites [99]. Dark chocolate with high cocoa content (>85%) has been shown to reduce NADPH oxidase activation, ROS production, and eicosanoid formation by platelets, thus improving arterial dilation, inhibiting LDL oxidation, and increasing HDL cholesterol [104]. Extra virgin olive oil (EVOO) also exerts antioxidant properties. It reduces postprandial markers of OS, including ROS, isoprostane 8-iso-PGF2α-III, NOX2 activity (the catalytic subunit of the NADPH oxidase enzyme), sE-selectin, and sVCAM1. Simultaneously, EVOO increases levels of vitamin E and decreases urinary markers of OS [104]. Finally, the administration of the probiotic *Lactobacillus paracasei* PS23 to aged mice was associated with a slowing and attenuation of muscle mass and strength decline, highlighting the potential role of gut microbiota in addressing sarcopenia [21].

The potential role and the recommended doses for the nutrients reviewed here are summarized in Table 1.

## 5. Physical Exercise in Patients with Stress Hyperglycemia or Diabetes

Physical inactivity has been shown to have detrimental effects on various conditions, including sarcopenia, aging, and diseases such as diabetes [107,108,109]. Age-related diseases, particularly those associated with prolonged physical inactivity or hospitalization, can accelerate the decline of skeletal muscle mass and physical function, thereby contributing to the onset of sarcopenia [110]. Exercise is widely recognized as a highly effective intervention against sarcopenia. Numerous studies have explored the impact of physical exercise programs on patients with sarcopenia and diabetes mellitus, demonstrating the potential benefits of exercise interventions in these populations.

### 5.1. Assessment of the Type of Exercise and Muscle Health

Most of the existing literature on the effects of exercise in patients with T2DM has fundamentally focused on aerobic exercise interventions. Aerobic exercise, such as walking or cycling, produces continuous, rhythmic movements of large muscle groups. This form of exercise is particularly effective in improving metabolic regulation and cardiovascular function and has been shown to combat both sarcopenia and T2DM [111]. Kirwan et al. demonstrated that aerobic activity in patients with T2DM significantly enhances glycemic control and insulin sensitivity, increases cardiac output, and is associated with substantial reductions in cardiovascular and overall mortality risk compared to sedentary patients [112]. Improvements in glycemic profile are observed by reductions in HbA_1c_, fasting plasma glucose, and IR [113,114,115]. Additionally, aerobic exercise also enhances skeletal muscle mass, particularly leg strength, making it highly recommended for older adults [116,117,118,119]. High-intensity interval training has emerged as a particularly effective and time-efficient form of aerobic exercise for patients with T2DM or sarcopenic obesity [116,117].

In the last two decades, resistance training has gained considerable attention as an alternative physical training option for patients with T2DM. Resistance exercise involves movements utilizing weights, bodyweight exercises, or elastic resistance bands. Like aerobic exercise, resistance training improves glycemic profiles and insulin sensitivity [120,121]. Castaneda et al. reported notable improvements in diabetes medication with resistance training but no changes in HbA_1c_ levels [122]. Resistance training has been found to significantly increase the levels of IGF-1, which enhances insulin sensitivity, stimulates muscle regeneration, and promotes improvements in muscle mass and strength [117,123].

Structured resistance exercise of either high or low intensity has been shown to effectively mitigate the severity of sarcopenia. This makes resistance training particularly suitable for frail, elderly individuals. Enhancements in muscle quality, strength, and skeletal muscle mass associated with resistance training reduce the risk of falls and improve physical performance, making it an ideal intervention for individuals with sarcopenia [113,124,125,126]. However, proper supervision is essential, as resistance training may increase the risk of musculoskeletal injuries in older adults [127].

In many studies, a combination of aerobic and resistance exercise has been employed to maximize the benefits of both modalities. This dual approach not only improves cardiovascular function and metabolic parameters but also achieves better glycemic control while increasing muscle mass, strength, and functional performance [117,125,128,129].

Beyond changes in muscle mass and results in physical performance tests, the impact of exercise on quality-of-life parameters has also been examined in one study. However, no significant differences have been observed in general health, psychological well-being, social relationships, or environmental domains. This lack of change has been attributed to the psychological distress, frustration, and feelings of powerlessness often experienced by individuals with diabetes [130].

In other studies, exercise programs were combined with nutritional interventions in patients with diabetes, demonstrating additional benefits, such as improved blood glucose levels, enhanced lipid metabolism, and reduced inflammation [131,132]. Nevertheless, other studies reported no significant improvements in glycemic control, the metabolic effects of such combined interventions remain inconsistent, and more research is needed [133,134].

While most reviews and meta-analyses recommend resistance training [121,135], combining resistance and aerobic exercise may offer the most comprehensive benefits for managing sarcopenia and T2DM [136,137].

### 5.2. Appropriateness of the Timing of Physical Exercise

The optimal timing and frequency of exercise for patients with T2DM and sarcopenia remain unclear. Most available evidence is derived from studies in which participants performed physical exercise three times a week, typically on nonconsecutive days [130,131,132,138].

In patients with sarcopenia without diabetes mellitus, exercising just before protein supplementation has traditionally been considered the ideal timing to maximize muscle response. However, recent studies in frail elderly individuals suggest that the muscle response remains effective regardless of the timing of exercise relative to supplementation [138].

The latest recommendations from the ADA regarding physical exercise in patients with diabetes mellitus are as follows [70]: (i) at least 150 min per week of moderate to vigorous aerobic activity. Ideally, each session should last at least 30 min and be performed 3 to 7 days per week; (ii) moderate to vigorous progressive resistance training 2 to 3 times per week on nonconsecutive days; and (iii) for older adults, flexibility and balance exercises are recommended 2 to 3 times per week to reduce the risk of falls and improve overall mobility. Figure 3 summarizes the benefits of resistance and aerobic exercise and offers recommendations on the periodicity of their practice.

Resistance training for older adults with frailty should be performed 2–3 times per week, starting at 20–30% of one repetition maximum (1 RM) and gradually progressing to 80% of 1 RM. Power exercises should be included at 30–60% of 1 RM to induce marked improvements in functional task performance. Functional and balance training, such as sit-to-stand exercises and tandem foot standing, enhances daily activity capacity and stability. Endurance training, including walking and stair climbing, should begin after strength and balance improvements, progressing from 5–10 min to 15–30 min at moderate intensity. Gradual progression in volume, intensity, and complexity is essential for long-term benefits [125].

Participation in supervised exercise programs is also advised to ensure safety and optimize the health benefits of physical activity in individuals with T2DM.

## 6. Genetic and Lifestyle Factors Affecting the Response to Nutrition and Physical Exercise

Several genetic and lifestyle factors can modulate the effectiveness of interventions designed to improve metabolic control, muscle mass and strength. These factors are especially important in patients with T2DM and sarcopenia.

Variants in genes related to insulin signaling, such as IRS-1 and PPARG (peroxisome proliferator-activated receptor gamma), have been shown to influence insulin resistance and BMI [139,140,141]. Other different polymorphisms have also been linked to IR [142]. Consequently, the presence of any of these may modify the effectiveness of dietary and physical exercise interventions. Some polymorphisms in genes such as ACTN3 and MSTN (myostatin) can inhibit muscle growth and affect the ability to develop strength and muscle mass [143,144]. These variants might influence the response to resistance exercises and nutritional interventions rich in protein. Other polymorphisms in genes related to inflammation, such as IL-6 and TNF-α and others, can modulate muscle regeneration capacity [145,146,147].

Some modifiable lifestyle habits can also modify the response to nutritional and exercise-related interventions. Smoking, for instance, is known to reduce muscle protein synthesis, increase oxidative stress, and aggravate insulin resistance [148,149,150,151]. Alcohol consumption also interferes with muscle regeneration and insulin signaling [152]. Similarly, chronic stress, sleep deprivation, and a pro-inflammatory diet accelerate the aging process and impact metabolic complications [153].

## 7. Conclusions

T2DM and sarcopenia are common and interrelated conditions that exist in a bidirectional relationship. Individuals with diabetes are at heightened risk for the development of sarcopenia and for progress. Sarcopenia, in turn, exacerbates IR by reducing tissue responsiveness to insulin. This increase in IR leads to hyperglycemia, further contributing to T2DM. This association is linked to several adverse outcomes, including premature mortality, increased hospitalization rates, and diminished functional capacity and quality of life. Hyperglycemia-induced glucotoxicity plays a key role in this cycle by activating glycosylation of proteins, generating OS, and promoting a state of chronic systemic inflammation. These processes collectively impair skeletal muscle by inhibiting protein synthesis, reducing oxidative capacity, and further aggravating IR—thereby reinforcing the vicious hyperglycemia–sarcopenia circle.

To address these challenges, hyper-proteic nutritional supplements tailored to individuals with diabetes are recommended. Supplements containing a high or exclusive proportion of soluble fiber may offer additional benefits. Formulations enriched with whey proteins rich in branched-chain amino acids could be particularly effective due to their potential to enhance muscle anabolism and improve glycemic control. Replacing carbohydrates with MUFA could also be useful for better diabetes management and could improve certain nutritional parameters, while the benefits of antioxidant nutrients and omega-3 fatty acids in this population remain inconclusive.

In conclusion, current evidence suggests that an integrated approach combining optimal nutrition with structured, supervised resistance and aerobic exercise—performed at least three times per week on non-consecutive days—is the most effective strategy for improving metabolic control and body composition in individuals with T2DM and sarcopenia. Nonetheless, further clinical trials are needed to better elucidate the effects of specific nutritional interventions in combination with targeted exercise programs.

## Figures and Tables

**Figure 1 nutrients-17-00499-f001:**
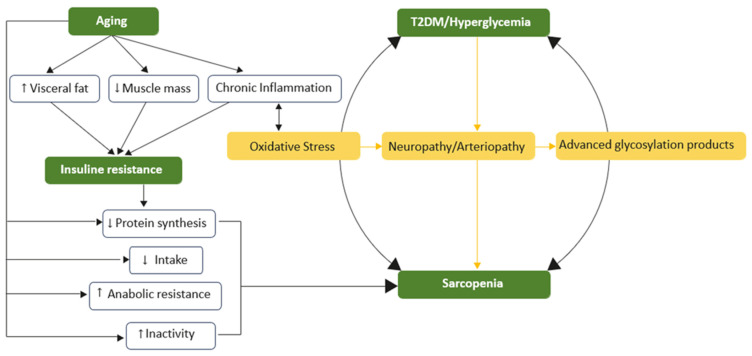
The vicious cycle of hyperglycemia and sarcopenia. Insulin resistance is the central and common element of aging that underlies the association between type 2 diabetes mellitus and muscle atrophy. Low skeletal muscle mass, visceral fat, and chronic inflammation contribute to impaired glucose uptake and elevated blood glucose levels. Aging also leads to sarcopenia through mechanisms such as reduced protein synthesis, heightened anabolic resistance, and increased physical inactivity. Chronic inflammation further exacerbates these issues, worsening oxidative stress, blood circulation, and glucose disposal.

**Figure 2 nutrients-17-00499-f002:**
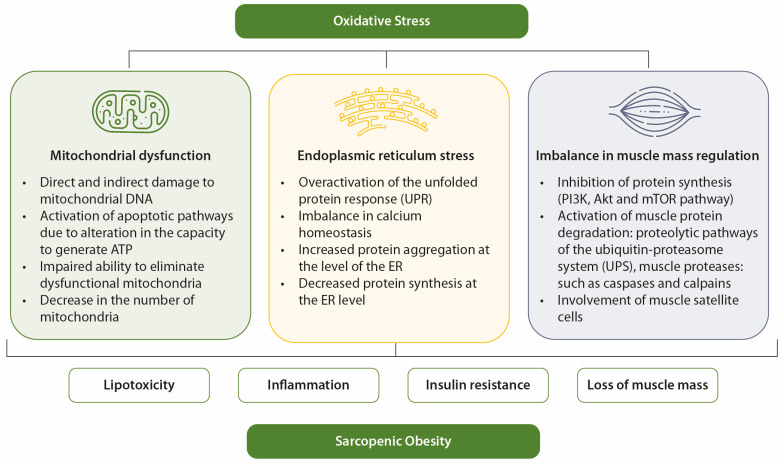
Mechanisms of OS in sarcopenic obesity. OS favors sarcopenia and obesity through mitochondrial dysfunction, ER stress, and imbalance in muscle mass control.

**Figure 3 nutrients-17-00499-f003:**
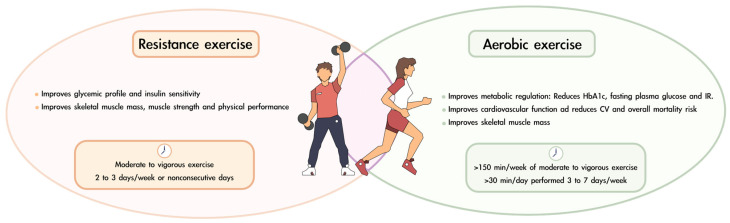
Benefits and recommendations for combining resistance and aerobic exercise in patients with diabetes and sarcopenia.

**Table 1 nutrients-17-00499-t001:** Nutrients with potential effects on sarcopenia and diabetes overlap.

Nutrient	Function	Recommended Dose
Total protein		1–1.5 g/kg body weight [5]
Whey protein	Promotion of muscle protein synthesis Improvement of muscle strength	20–40 g per day [105]
Leucine	Increased ability to recover postprandial muscle protein synthesis	2–3 g per day [84]
Arginine	Increase lean body mass in the short term. Provide amino acid balance and increase HMB–Leu activity in the treatments used	11–15 g/day [84]
Glutamine	Provide amino acid balance and increase HMB–Leu activity in the treatments used	14 g/day [84]
Vitamin D	Suppression of the activity of transcription factors associated with muscle atrophy. Stimulates muscle protein synthesis via mTORC1. Improvement of mitochondrial function and reduction of skeletal muscle fat accumulation. Synergy with leucine to enhance muscle anabolism.	20–25 mcg/day (800–1000 UI/day)
Omega-3	Anti-inflammatory effects Enhancement of Muscle Protein Synthesis Prevention of Muscle Breakdown	>2.5 g/day [106]

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
