# Peer review of "Medical Nutrition Therapy and Physical Exercise for Acute and Chronic Hyperglycemic Patients with Sarcopenia"

_nutrients, 2025, doi:10.3390/nu17030499_

Round 1

Reviewer 1 Report

Comments and Suggestions for Authors

Ángel Luis Abad-González et al presented a manuscript titled: “Medical nutrition therapy and physical exercise for acute and chronic hyperglycemic patients with sarcopenia”. The article is a comprehensive review of the interplay between hyperglycemia, sarcopenia, and the potential benefits of medical nutrition therapy and physical exercise. It highlights the novel concept of a “vicious cycle” linking hyperglycemia and sarcopenia, emphasizing the role of insulin resistance as a central element. The discussion on specific nutritional interventions and exercise regimens tailored for hyperglycemic patients with sarcopenia is a rather valuable contribution to the field. However, there are still areas of the manuscript with ambiguities and flaws which need to be addressed:

1. While you identified insulin resistance as a key factor, your manuscript lacks a detailed exploration of the underlying biochemical and molecular mechanisms. A deeper dive into these pathways could enhance understanding and provide clearer targets for possible future interventions.

2. You did not adequately address how individual differences (for example: genetic factors, lifestyle variations) might affect the response to nutritional and exercise interventions.

3.. Furthermore, although you suggested specific dietary components (such as omega-3 fatty acids, whey proteins), you still lack clarity on the optimal dosages and combinations for different patient profiles.

4. Your recommendations for exercise are somewhat generic. More precise guidelines tailored to different levels of sarcopenia severity or patient capabilities would be beneficial.

5. The acknowledgment of medical writing services sponsored by Fresenius Kabi raises questions about potential biases in favor of certain nutritional products or strategies.

Reviewer 2 Report

Comments and Suggestions for Authors

To the Authors

The study has addressed the potential relationship between metabolic imbalances, IR, and sarcopenia in diabetic patients. The review conclusions suggest that T2DM and sarcopenia are common conditions linked in a bidirectional relationship. Additionally, the study identifies the potential lifestyle and nutritional interventions that may help protect against sarcopenia in diabetic patients. In summary, the study sheds light on the potential association between metabolic syndrome and sarcopenia, which could lead to the development of new treatment options.

To improve the clarity of the article I suggest some minor revisions:

·       130 This decrease leads to an increased IR – reformulated as ‘This decrease contributes to an increased IR’ because the affirmation is misleading, and more complex mechanisms are involved.

·       Line 143-144 - In fact, chronic systemic inflammation is the main factor influencing the pathophysiology of sarcopenic obesity. – rephrase this affirmation, because the explanation for this is provided in the subsequent sections.

·       Please provide more figures.

·       Please check the whole manuscript for typos, missing points, and commas as well as for unnecessary gaps.

Round 2

Reviewer 1 Report

Comments and Suggestions for Authors

The authors have significantly improved the quality of their manuscript and it is now fit for publication.

Author Response

Thank you very much for taking the time to review this manuscript.
